# Plywood Made from Plasma-Treated Veneers: Investigation of Performance Differences between Plasma-Pretreated and Untreated Beech Veneers at Comparable Melamine Resin Load

**Richard Wascher [1,\*], Georg Avramidis [1] and Wolfgang Viöl [1,2]**

[1] Faculty of Engineering and Health, University of Applied Sciences and Arts, Von-Ossietzky-Strasse 99, 37085 Göttingen, Germany; georg.avramidis@hawk.de (G.A.); wolfgang.vioel@hawk.de (W.V.)

[2] Application Center for Plasma and Photonics, Fraunhofer Institute for Surface Engineering and Thin Films, Von-Ossietzky-Strasse 100, 37085 Göttingen, Germany

\* Correspondence: richard.wascher@hawk.de; Tel.: +49-551-3705293

**Abstract:** In this study, the dimensional stability and mechanical properties of plywood made from untreated and plasma-pretreated beech veneers were compared. The wood veneers used (native and thermally modified) were impregnated with melamine resin in a simple dipping process prior to plywood production. The duration of the impregnation process was adjusted to give the same melamine resin loading for the different veneer types, with the plasma-pretreated veneers requiring only a fraction of the impregnation time compared with non-plasma-pretreated veneers. With comparable melamine loading, testing of the mechanical properties of the plywood for the different specimen collectives showed significant differences in some cases with respect to compressive strength, bending strength and tensile strength (with the associated moduli of elasticity). For example, it was shown that plywood made from plasma-pretreated native beech veneers shows an increase in bending strength of about 8%, and from plasma-pretreated and thermally modified beech veneers, there is an increase of about 10% compared to the reference.

**Keywords:** plasma treatment; dielectric barrier discharge; dimensional stability; liquid uptake; mechanical properties; beech veneers; thermal modification; melamine resin

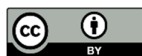

## 1. Introduction

The renewable resource wood has served humans as a construction material for thousands of years and is still the subject of numerous scientific publications due to its manifold advantages, such as $CO_2$ neutrality, haptics, physical properties, etc. [1–3]. However, wood as a structural material is plagued by a number of crucial drawbacks that significantly restrict the application breadth of this material. For example, the swelling and shrinkage behavior, as well as the susceptibility to biological degradation, represent a constant challenge for both users and researchers who are concerned with optimizing this material [1,2]. Conventional measures to overcome this problem are, for example, constructive solutions or sealing the surfaces with suitable coatings. Modern methods are based on chemical or physical–chemical modification of the material (acylation, furfurylation, thermal modification, etc.), which has led to technological advances and successful commercialization of some of these methods [1]. The mode of action is usually based on blocking hydrogen bonds or preventing water penetration into the wood structure [1–3], which consequently leads to reduced swelling of the wood and thus to an improvement in dimensional stability. Especially in the case of chemical modification, the amount of modifier deposited in the wood cell walls is the decisive factor for the dimension-stabilizing effect [4,5]. Typically, vacuum-pressure methods are used for impregnating wood with the modifiers, since simple dipping processes produce only deficient results. This is

due to the low pressure differences (ambient pressure/pressure in the capillaries), which make the capillary forces insufficient. Vacuum-pressure methods are time-consuming and require a relatively large amount of equipment; however, they ensure extensive impregnation of the modifiers in the wood volume. Still, vacuum-pressure methods are time-consuming and require a relatively large amount of equipment, even though they ensure thorough impregnation of the wood volume with modifiers [1].

Studies demonstrate that the use of technical plasmas as a pretreatment step in combination with simple immersion processes for wood veneers leads to a high permeation of the wood structure with modifier [6]. In this context, wood veneers were exposed to a gas discharge, using oxygen-containing gases (air), which resulted in higher permeation of the substrate and, more generally, improved uptake of modifiers and coating agents [7–11]. One explanation may be found in the hypothesis postulated by Wascher et al. that under specific conditions micro-discharges can occur in the cavities of thin wood veneers. The authors derived this hypothesis from the observation that the typical plasma effects (decrease in surface energy and pH) could be detected not only in the near-surface areas, but also in the core material of plasma-treated wood veneers [12]. As a consequence, plywood samples made from beech veneers treated with plasma and subsequently impregnated with melamine resin showed higher shear strength than the reference [13]. Recent studies [9,13,14] on the comparability of plywood samples made from plasma-pretreated and non-pretreated beech veneers raise the following question: how do plywood specimens perform with respect to dimensional stability and mechanical properties when the individual (non-plasma-pretreated) veneers of these specimens have the same melamine resin uptake—with a correspondingly longer immersion time—as the plywood made from plasma-pretreated veneers (impregnation time 1 s)? Although this question has already been briefly touched upon in Reference [13], there is a lack of systematic investigation and discussion on this issue. The present study aims to close this gap of knowledge.

## 2. Materials and Methods

In order to provide a concrete reference to the previously discussed and published studies [9,13], native and thermally modified beech veneers and melamine resin were used in this study. In the context of this study, the term "reference" is used exclusively for the plywood specimens made from non-plasma-pretreated beech veneers, where these veneers remained in the immersion bath until a liquid uptake or WPG (WPG = Weight Percentage Gain) of plasma-pretreated veneers (1 s impregnation time) was achieved. The test matrix for determining the impregnation time required for the reference is depicted in Figure 1. In fact, only comparisons of the plasma-pretreated samples and the reference with the same load will be treated in this study, without including in the tests the references that have the same impregnation time (1 s, with a correspondingly lower load) as the plasma-pretreated samples. The comparisons of the plasma-treated samples with the "regular" references (in the proper sense), where the relevant process parameters (impregnation time of the veneers) remained the same as the parameters of the plasma-treated samples, have already been discussed in detail [9,13].

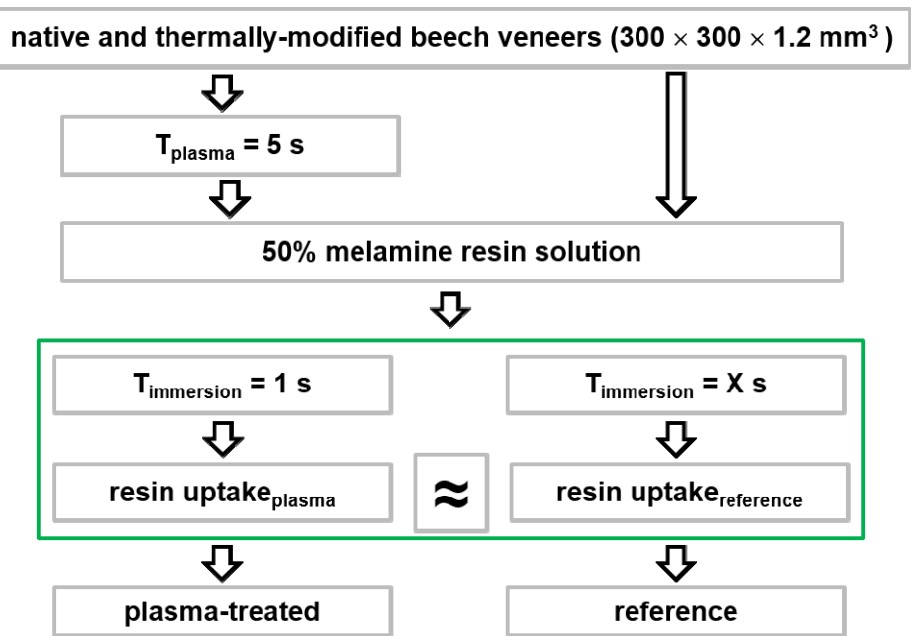

**Figure 1.** Schematic illustration of the test matrix.

Native and thermally modified rotary cut beech (*Fagus sylvatica*) veneers with a thickness of 1.2 mm were purchased from OWI GmbH, Germany. Thermally modified veneers with a residual moisture content of 12% were manufactured at a treatment temperature of about 200 °C and an exposure time of 20 min [15]. For subsequent experiments, the samples were climatized at standard conditions (20 °C and 65% RH) for 24 h, resulting in an equilibrium moisture content (EMC) of about 10% for unmodified beech and 7.7% for thermally modified beech. Melamine resin (Madurit MW 840; solid content, 75%; pH value, 9–11; and specific gravity, 1.245–1.260 g/mL) was purchased from INEOS Group AG, Germany.

Figure 2 shows the dielectric barrier discharge (DBD) setup used for pretreatment of the veneers. The veneer sheets were transported centrally by a roller conveyor through the gap (5 mm) between two electrode arrays. Each electrode array consisted of five individual electrodes made of rectangular $Al_2O_3$ tubes ($15 \times 15 \times 400$ mm$^3$; wall thickness = 2.5 mm) filled with bronze powder, with a gap of 5 mm between the individual electrodes of an array [9,13,16]. Two radial fans above and below the electrode arrays generated an ambient air flow towards the electrode arrays (cooling and discharge homogenization). The upper electrode array was connected to a high voltage with alternating polarity and a pulse repetition frequency of 15 kHz (pulse duration = 2 µs) the lower electrode array was grounded. The applied alternating voltage was ~18 kV (peak), leading to a power density of 3.5 W/cm$^2$ for beech veneers and 3.7 W/cm$^2$ for thermally modified beech veneers. During plasma treatment, gas temperatures were measured with a fiber optic thermometer (FTI-10, FISO Technologies, Quebec, Canada) and did not exceed 60 °C.

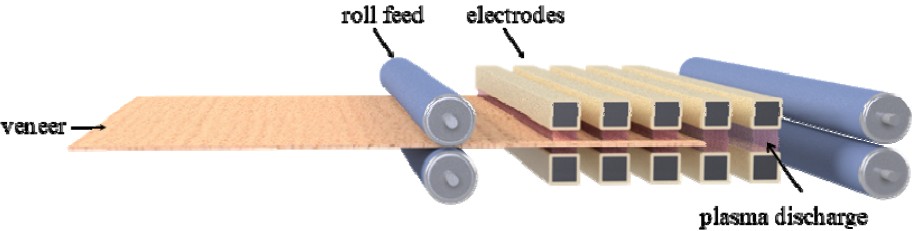

**Figure 2.** Plasma setup.

The liquid uptake of the specimens was measured directly after plasma treatment in a melamine resin impregnation bath at atmospheric pressure and a liquid temperature of 20 °C. To determine the resin uptake, veneers with dimensions $300 \times 300 \times 1.2$ mm³ were completely immersed in the impregnation bath. The residual liquid was removed from the veneer surface, using a stripping device. The stripping device as a self-construction consists of two rubber lips lying on top of each other. The residual liquid is removed by passing an impregnated veneer through the two rubber lips. All samples were weighed before plasma treatment (digital balance SSH93, SCALTEC Instruments GmbH, Heiligenstadt, Germany). After immersion and removal of the residue from the surface, the samples were weighed and the liquid uptake was calculated according to the following equation [1,17]:

$$liquid\ uptake\ [\%] = \frac{M_2 - M_1}{M_1} * 100, \tag{1}$$

where $M_1$ is the mass before immersion (at corresponding EMC), and $M_2$ is the mass after immersion. In order to determine the solid content of the melamine resin remaining in the material after impregnation, the WPG (Weight Percentage Gain) was calculated by using the following equation:

$$WPG\ [\%] = \frac{M_m - M_u}{M_u} * 100, \tag{2}$$

where $M_m$ is the weight of the oven-dried immersed samples, and $M_u$ is the weight of the oven-dried untreated samples [1].

For the production of plywood, veneer sheets were impregnated in a 50% melamine resin solution. The impregnated veneers were dried for 24 h and then pressed by using a hot press (LAP 40 Gottfried Joos Maschinenfabrik GmbH & Co. KG, Pfalzgrafenweiler, Germany, pressing pressure 25 kg/m², pressing time 3 min/mm, temperature 130–140 °C) into crosswise (90°) oriented five-layer panels. No additional glue was applied between the single veneer sheets—in accordance with the standard manufacturing process of plywood from thermally modified beech veneers at OWI GmbH, Lohr am Main, Germany, which saves one process step in this way. The glue line produced by the melamine resin exclusively available in the impregnated veneers creates the bonding between the individual veneers. Changes in the outer dimensions of a wood sample during a swelling–drying process provide information about its dimensional stability. Quantitatively, an increase or decrease in the dimensional stability of the sample can be represented by the swelling coefficient S:

$$S\ [\%] = \frac{V_{WS} - V_{OD}}{V_{OD}} * 100, \tag{3}$$

where $V_{WS}$ is the water-swollen volume of the sample (WS), and $V_{OD}$ is the oven-dry volume of the sample (OD) [1]. When calculating the swelling coefficient of plywood, only the radial direction (thickness) was used, since tangential and longitudinal directions cannot be measured separately due to the typical 90° orientation of the veneer layers [18]. For this reason, the equation was adapted accordingly to the following:

$$S^*\ [\%] = \frac{l_{WS} - l_{OD}}{l_{OD}} * 100, \tag{4}$$

where $l_{WS}$ is the water-swollen thickness of the sample, and $l_{OD}$ is the oven-dry thickness of the sample. To measure the thickness (sample dimension $25 \times 25$ mm²) a dial indicator was used.

The tensile strength (TS) and the corresponding modulus of elasticity (MOE) were determined according to DIN 52 377 [19] ($n = 10$). However, the lateral dimensions of the tensile bars were reduced by a factor of 2.3 due to the maximum treatment width (300 mm) of the plasma system used (total length according to DIN 52 377: 400 mm). Bending tests according to DIN EN 310 [20] were performed to determine the bending strength or

modulus of rupture (MOR) and the corresponding modulus of elasticity (MOE) ($n = 10$). The compressive strength (CS) was determined according to DIN 52 376 [21] ($n = 20$). Two types of plywood panels were tested for each test parameter: specimens manufactured with the grain of the outer layers parallel and perpendicular to the direction of force. The tests according to DIN 52 377, DIN EN 310 and DIN 52 376 were performed on a 100 kN universal testing machine (Zwick GmbH & Co. KG, Germany) and analyzed with the according software TEST EXPERT II (Zwick GmbH & Co. KG, Germany).

### 3. Results and Discussion

To ensure comparable resin uptake (see Figure 1), the uptake values were determined separately for the plasma-treated samples of native and thermally modified veneers after an impregnation time of 1 s. This value will be referred to as the "target value" in the following. Subsequently, veneers (native and thermally modified) were impregnated without plasma pretreatment until the respective target value was achieved, depending on the veneer type. Figure 3 depicts the melamine resin uptake, as well as the WPG. The impregnation time required for the reference to reach the value of melamine resin uptake or WPG of plasma-pretreated veneers after 1 s impregnation time was approximately 90 s for native beech veneers and approximately 900 s for thermally modified beech veneers. These impregnation times were used in the following to impregnate the wood veneers.

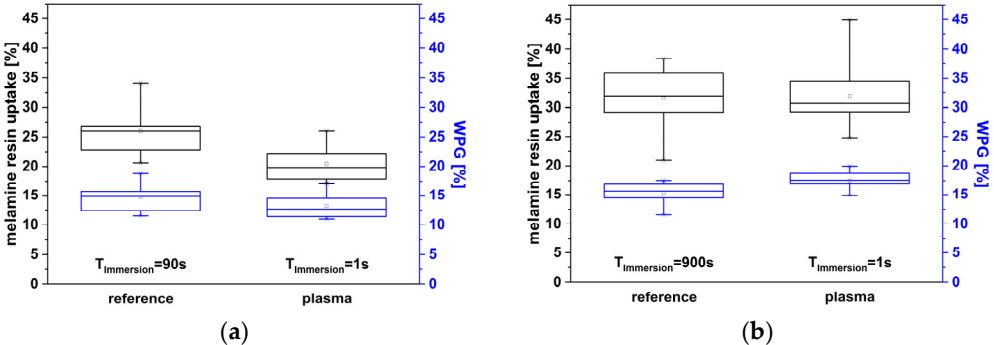

**Figure 3.** Melamine resin uptake and WPG of native (**a**) and thermally modified (**b**) beech veneers. Error bars display the standard deviation.

Subsequently, five-layer plywood panels were made from the impregnated veneers. The plywood was manufactured without an additional glue joint, with only the melamine resin deposited in the veneers acting as an adhesive [9,13].

The density determination of the plywood showed approximately 824 kg/m³ for reference and approximately 834 kg/m³ for plasma for plywood made of native beech veneers and approximately 793 kg/m³ for reference and approximately 807 kg/m³ for plywood made of thermally modified beech veneers. A two-sample t-test showed that the respective pairs of samples were not significantly different from each other. The density measurement supports the statement (see Figure 3a,b) of a comparable melamine resin load in the individual veneers after plywood production.

The plywood was subsequently tested with regard to dimensional stability (Figure 4).

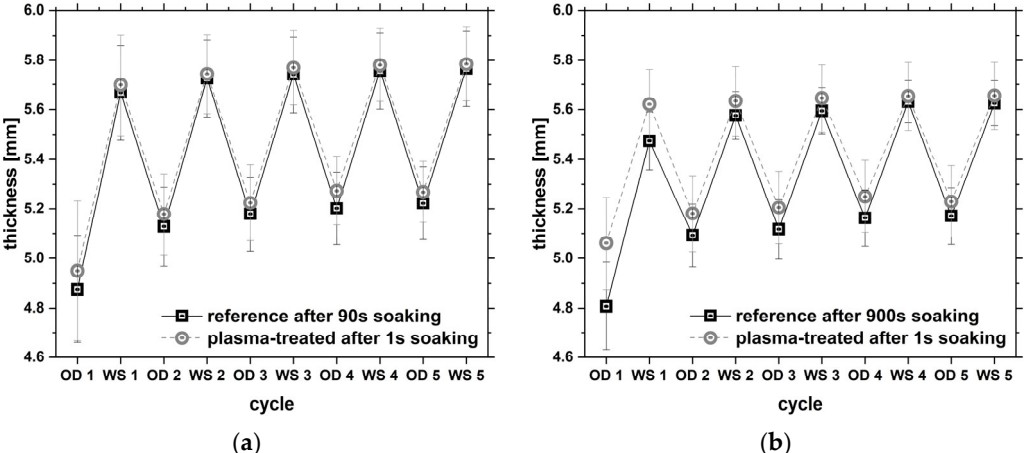

**Figure 4.** Changes in thickness of plywood made of native (**a**) and thermally modified (**b**) beech veneers in relation to series of cycles for determining dimensional stability. Error bars display the standard deviation.

In the case of the plywood specimens made from native beech veneers (Figure 4a), no significant differences in thickness swelling were observed between the reference and the specimens made from plasma-pretreated veneers. In addition, the initial thickness was almost the same between the two sample collectives.

For the plywood specimens made from thermally modified beech veneers (Figure 4b), a difference in terms of thickness swelling as a result of the wet–dry cycling between the reference and the specimens made from plasma-pretreated veneers was observed. In contrast to the samples made from native beech veneers (Figure 4a), the two sample collectives differed significantly in initial thickness. The specimens made from plasma-pretreated veneers obviously showed higher dimensional stability than the reference specimens when undergoing wet–dry cycles.

These observations (differences in initial thickness and dimensional stability) can be explained by the fact that, despite the approximately equal amount of melamine resin absorbed by the two sample collectives (reference and plasma-pretreated), there was a different distribution and penetration of the modifier reagent within the wood structure. The different distribution or penetration depth of the modifier within the wood structure in the case of the plywood from plasma-pretreated veneers may have implications when the veneers are pressed into the plywood. Deeper penetration of the modifier and deposition of the same within the cell walls of the wood results in higher incompressibility of the material, so that the plywood from plasma-pretreated veneers may have a greater material thickness.

The swelling of the wood is essentially caused by water incorporation between the fibrils within the wood cell wall [22,23]. Due to the deposition of modification reagents (here: melamine resin) into the wood cell wall [5,24], the incorporation of water into the cell wall is blocked or reduced. In the plasma-pretreated veneers, the melamine resin was probably incorporated within the cell wall and the vessel elements in the deeper layers of the wood structure. This results in a higher volume of wood that can interact with the melamine resin, so that the swelling of the wood is reduced compared to the non-plasma-pretreated samples.

The hypothesis postulated in Wascher et al. [12] offers a possible explanation for the observation and the subsequent assumption that the melamine resin is more deeply incorporated in the plasma-treated samples. The authors claim that if the electric field is large enough when applying a high voltage to thin and porous materials such as wood veneers in a volume discharge, micro-discharges can be generated in the cavities (e.g., cracks and vessel elements) of a wood structure and known plasma effects (generation of

polar groups, acidification, etc.) can be generated on the cavity surfaces (e.g., vessel walls). Due to the generation of these plasma-induced surface effects inside the wood veneers, the surfaces of the cavities have a higher surface energy than the cavity surface of the untreated veneers, which means that the flow rate of the modifier inside the wood structure can be increased and thus the infiltration of the material is accelerated in plasma-treated samples.

Considering the simplified Washburn equation based on Hagen–Poiseuille's law and the Young–Laplace equation [25], which describes the penetration of a liquid into a porous material, a direct correlation is obtained between an increase of the surface energy on the surfaces of the cavities inside the wood after plasma treatment and the flow rate of the melamine resin.

$$h = \sqrt{\frac{D * \gamma * t * cos\phi}{4 * \eta}} \qquad (5)$$

With the time, $t$, a liquid with a dynamic viscosity, $\eta$, and a surface tension, $\gamma$, penetrates by the height, $h$, into the capillary, whose pore diameter is $D$. Here, $\phi$ is the contact angle between the penetrating liquid and the solid (cavity wall).

According to this equation, the liquid uptake and spreading of the modifier within the material occurs faster if the surface energy (derived from the contact angle $\phi$) of the areas of the wood structure affected for infiltration is increased (this means low $\phi$), e.g., by plasma pretreatment. Taking the above reasoning into account, it can be assumed that increased infiltration of the modifier in the wood bulk (at the same uptake quantity) takes place in the plasma-pretreated samples.

In considering the flow model in this way, the distribution of the melamine resin in the reference samples (generated by keeping the samples in the soaking bath for a longer time, 90 s for native and 900 s for thermally modified beech veneers) should be mainly confined to layers near the surface of the wood veneers. In contrast, the same amount of melamine resin is transported to the deeper layers in the plasma-treated samples (1 s soaking time) due to the plasma-induced increase in surface energy of the affected areas inside the wood (e.g., cavity walls). Another effect of the plasma treatment could be an increased diffusion of the melamine resin into the cell walls [14].

It is known that the degree of plasma-induced effects depends on the wood species or on the type of modification of the wood (e.g., thermal modification) [8,9]. This explains the differences in terms of hydrophilization of the cavities inside the wood achieved by plasma treatment between the two types of beech veneer used (native and thermally modified). In order to investigate a possible influence on the differences regarding the infiltration of the samples as a result of plasma pretreatment—compared to non-plasma pretreated samples—tensile strength and flexural strength with the corresponding E-moduli, as well as compressive strength, were tested, taking into account the fiber orientation. The results of these tests are presented in the form of a radar diagram (all values in tabular form can be found in the Appendixes A and B) to visualize differences (Figures 5 and 6). What is immediately apparent from the two figures is that (marginal) differences exist in all the strength values examined despite the same loading with the modification reagent.

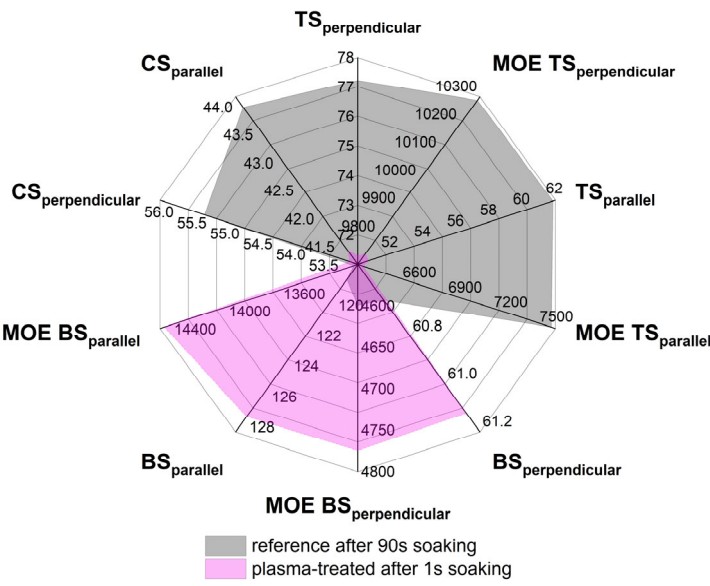

**Figure 5.** Radar diagram regarding mechanical properties of plywood made from untreated and plasma-treated native beech veneers.

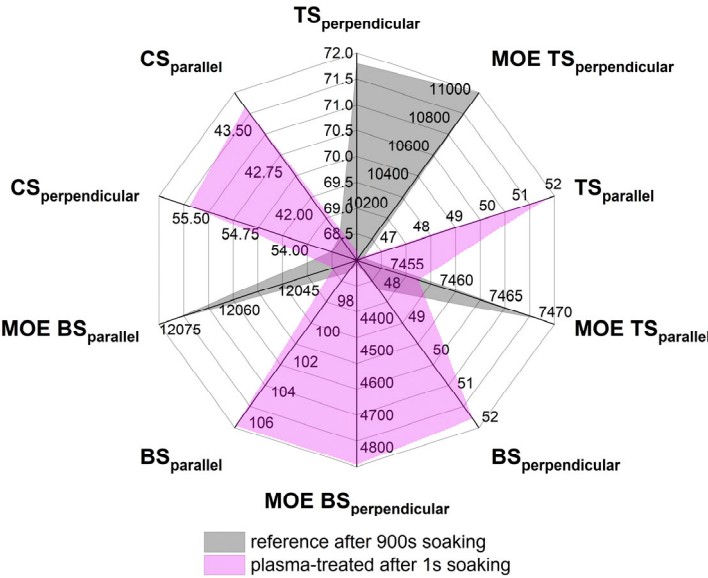

**Figure 6.** Radar diagram regarding mechanical properties of plywood made from untreated and plasma-treated thermally modified beech veneers.

In the plywood samples made from native beech veneers (Figure 5), the reference shows higher strength values in compressive strength (CS), as well as tensile strength (TS), with the associated E-moduli (MOE) both parallel and perpendicular to fibers, with significant differences in tensile strength and the respective E-modulus parallel to fibers. The values of bending strength (BS) with the corresponding E-moduli in both fiber directions are higher for the plywood made from plasma-pretreated beech veneers than for the reference, with only the E-modulus parallel to the fibers differing significantly.

In the case of the plywood samples made from thermally modified veneers, the graph shows higher strength values for TS (parallel to the fiber), BS and MOE BS (perpendicular to the fiber), BS (parallel to the fiber) and CS in both fiber directions. Only for tensile strength and MOE TS (measured perpendicular to the fibers), MOE TS (parallel to the

fiber), and MOE BS (parallel to the fiber) does the reference show higher strength values than the plywood made from plasma-treated veneers.

The compressive strength of melamine resin-impregnated wood may be increased due to the incorporation and crosslinking of the modifier in the cell wall [26]. In contrast, tensile strength behaves differently: due to the incorporation of modifier in the cell wall and resulting crosslinking, the natural elasticity of wood fibers in chemically modified wood may decrease and the frictional resistance between fibers may increase [27].

Bending strength represents a combination between tensile and compressive strength. In plywood, the maximum stresses under bending loads occur in the outer veneer plies and nearly zero in the middle veneer ply [17].

In terms of bending strength, plywood samples from both native and thermally modified beech veneers show an increase after plasma pretreatment compared to references with the same melamine resin loading (Figure 3a,b). The tensile and bending strength values, as well as the corresponding E-moduli, behave rather erratically for specimens from native and thermally modified beech veneers with and without plasma pretreatment. While the compressive strength values of the plywood from thermally modified beech veneers are higher after plasma pretreatment than the corresponding reference, the case is reversed for the specimens from native beech veneers.

## 4. Conclusions

In this study, it was demonstrated that, for simple immersion processes, plasma pretreatment of wood veneers accelerates the penetration of liquids into the wood core up to 900 times, depending on the materials used. Plasma-induced differences in penetration and infiltration behavior show an influence on the mechanical properties of plywood manufactured from plasma-pretreated and subsequently impregnated wood veneers. For example, it was shown that plywood made from plasma-pretreated native beech veneers shows an increase in bending strength of about 8% compared to the reference. Similar behavior in terms of mechanical properties is also expected for plasma pretreatment of composite (e.g., veneer/natural fiber) source materials, and this is being tested in ongoing studies.

**Author Contributions:** R.W. and G.A. designed the experiments; R.W. and G.A. performed the experiments; R.W. and G.A. analyzed the data; W.V. contributed reagents/materials/analysis tools; R.W., G.A. and W.V. wrote the paper. All authors have read and agreed to the published version of the manuscript.

**Funding:** This research was funded by the German Federal Ministry for Economic Affairs and Energy (grant number 03THWNI004).

**Acknowledgments:** The authors would like to thank Roger Skarsten from the HAWK for proofreading.

**Conflicts of Interest:** The authors declare no conflict of interest.

## Appendix A

**Table 1**. Data from plywood made from native beech veneers.

| | density | MOR | | MOE$_{MOR}$ | | TS | | MOE$_{TS}$ | | CS | |
|---|---|---|---|---|---|---|---|---|---|---|---|
| | kg/m³ (n=20) | N/mm² (n=10) | | N/mm² (n=10) | | N/mm² (n=10) | | N/mm² (n=10) | | N/mm² (n=20) | |
| | | parallel | perpendicular | parallel | perpendicular | parallel | perpendicular | parallel | perpendicular | parallel | perpendicular |
| reference | 823.8 ± 27 | 119 ± 12 | 60.7 ± 3 | 13230 ± 1116 | 4599.1 ± 254 | 61.8 ± 8 | 77.2 ± 12 | 7472.8 ± 657 | 10287.4 ± 847 | 43.8 ± 4 | 55.3 ± 5 |
| plasma | 834 ± 31* | 128 ± 8* | 61.1 ± 4* | 14619.1 ± 855 | 4774.4 ± 337* | 50.7 ± 8 | 71.3 ± 12* | 6236.4 ± 638 | 9736.4 ± 952* | 41.3 ± 2* | 53.1 ± 5* |

*The values of "plasma" are not significantly different (performed with two-sample test) from "reference".

## Appendix B

**Table 2**. Data from plywood made from thermally-modified beech veneers.

| | density kg/m³ (n=20) | MOR N/mm² (n=10) | | MOE$_{MOR}$ N/mm² (n=10) | | TS N/mm² (n=10) | | MOE$_{TS}$ N/mm² (n=10) | | CS N/mm² (n=20) | |
|---|---|---|---|---|---|---|---|---|---|---|---|
| | | parallel | perpendicular | parallel | perpendicular | parallel | perpendicular | parallel | perpendicular | parallel | perpendicular |
| reference | 792.5 ± 31 | 96.9 ± 6 | 48 ± 4 | 12076 ± 504 | 4230.2 ± 218 | 46.2 ± 4 | 71.8 ± 5 | 7468.2 ±619 | 11046 ±1671 | 41.4 ± 3 | 53.4 ± 2 |
| plasma | 807.1 ± 30* | 106.9 ± 8 | 51.7 ± 4 | 12035.7 ± 1110* | 4853.2 ± 360 | 51.8 ± 7* | 68.2 ± 10* | 7456.2 ± 681* | 9885.2 ± 1139* | 43.7 ± 1* | 55.5 ± 4* |

*The values of "plasma" are not significantly different (performed with two-sample test) from "reference".

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
