# Peer review of "Plywood Made from Plasma-Treated Veneers: Investigation of Performance Differences between Plasma-Pretreated and Untreated Beech Veneers at Comparable Melamine Resin Load"

_forests, doi:10.3390/f12101423_

Round 1

Reviewer 1 Report

The manuscript deals with the investigation and evaluation of the dimensional stability and mechanical properties of plywood fabricated from untreated and plasma-pretreated beech veneers. Please, see below my comments on your work:

In general, the abstract (lines 11 to 19) and the keywords (lines 20-21) correspond to the title, aims and objectives of the manuscript. The abstract is well-written and informative, and contains the main findings of the article. I’d recommend to include some specific results (values) obtained for the mechanical properties of the tested plywood samples.

In the keywords, I’d recommend to use the full term of DBD, i.e. “dielectric barrier discharge”. In addition, "load" is too general and can be replaced by another keyword or deleted.

The Introduction part is well written and informative, and provides sufficient information and references on the research topic.

In line 73, please add the full term, i.e. Weight Percentage Gain, followed by the abbreviation WPG.

In line 91, in addition to the solids content of the melamine resin used, please provide other relevant characteristics. Please explain why did you chose melamine resin as a bonding agent.

Please explain the choice of the selected parameters for veneer pre-treatment, i.e. frequency of 15 Hz and pulse duration 2 µs.

In line 112, please add some more information about the “stripping device”.

In lines 124-128, please justify the selected parameters of plywood hot pressing, i.e. pressing temperature, press time and pressure. Please add the pressure value in MPa.

In lines 147 – 155, please add the respective standards to the References of your manuscript.  

Overall, the Materials and Methods section is very well written and detailed.

In lines 169, the text in Figure 3 is a bit hard to read, please provide more clear graphs, if possible.

In lines 229-237, equation (5) and the related explanations do not belong to the Results and Discussion section, I’d recommend to move them to Materials and Methods.

In general, the results of the study are detailed, informative and properly discussed with relevant research works in the field.

The Conclusion part (lines 299-309) is specific and reflects the main findings of the manuscript.

The References cited are appropriate to the topic of the manuscript. Some of the references are not formatted in accordance with the journal requirements, please check the Instructions for authors.

Best regards!

Author Response

Responses to the Reviewer 1 comments and suggestions for corrections

Dear Reviewer, dear Editor,

Thank you very much for the valuable and helpful comments on our submitted manuscript. All changes in the manuscript are highlighted using the "Track Changes" feature.

Reviewer #1: 1) In general, the abstract (lines 11 to 19) and the keywords (lines 20-21) correspond to the title, aims and objectives of the manuscript. The abstract is well-written and informative, and contains the main findings of the article. I’d recommend to include some specific results (values) obtained for the mechanical properties of the tested plywood samples.

Authors: We have inserted a sentence in the text showing an example from the mechanical properties: “For example, it was shown that plywood made from plasma-pretreated native beech veneers shows an increase in bending strength of about 8%, and from plasma-pretreated and thermally-modified beech veneers an increase of about 10% compared to the reference.”

Reviewer #1: 2) In the keywords, I’d recommend to use the full term of DBD, i.e. “dielectric barrier discharge”. In addition, "load" is too general and can be replaced by another keyword or deleted.

Authors: Thank you for your suggestions. We have replaced the abbreviation "DBD" with "dielectric barrier discharge" and deleted the word "load".

Reviewer #1: 3) In line 73, please add the full term, i.e. Weight Percentage Gain, followed by the abbreviation WPG.

Authors: The abbreviation “WPG” was changed accordingly.

Reviewer #1: 4) In line 91, in addition to the solids content of the melamine resin used, please provide other relevant characteristics. Please explain why did you chose melamine resin as a bonding agent.

Authors: We added pH value and the specific gravity of the melamine resin. In the chapters "Introduction" and "Materials and Methods" we wrote (lines 68-78) that the current publication has a direct relation to the two publications of ours mentioned above. Therefore, all parameters, including the plasma-relevant settings, were kept the same in order to create a direct comparison. In addition, melamine resin is often used alongside phenolic resins for the production of veneer materials for exterior applications.

Reviewer #1: 5) Please explain the choice of the selected parameters for veneer pre-treatment, i.e. frequency of 15 kHz and pulse duration 2 µs.

Authors: In the chapters "Introduction" and "Materials and Methods" we wrote (lines 68-78) that the current publication has a direct relation to the two publications of ours mentioned above. Therefore, all parameters, including the plasma-relevant settings, were kept the same in order to create a direct comparison. Moreover, these are typical operating parameters for a DBD.

Reviewer #1: 6) In line 112, please add some more information about the “stripping device”.

Authors: We have inserted a sentence in the text “The stripping device as a self-construction consists of two rubber lips lying on top of each other. The residual liquid is removed by passing an impregnated veneer through the two rubber lips.”

Reviewer #1: 7) In lines 124-128, please justify the selected parameters of plywood hot pressing, i.e. pressing temperature, press time and pressure. Please add the pressure value in MPa.

Authors: As mentioned above, the present publication is directly related to the two publications of ours mentioned above. Therefore, all parameters were kept the same to allow a direct comparison. For this reason, we would also be reluctant to change physical units and parameters in order to maintain the recognition value.

Reviewer #1: 8) In lines 147 – 155, please add the respective standards to the References of your manuscript.

Authors: The literature references have been added.

Reviewer #1: 9) In lines 169, the text in Figure 3 is a bit hard to read, please provide more clear graphs, if possible.

Authors: We have scaled up the text in the two graphs accordingly.

Reviewer #1: In lines 229-237, equation (5) and the related explanations do not belong to the Results and Discussion section, I’d recommend to move them to Materials and Methods.

Authors: We agree with you in principle. However, we use this formula within the Results and Discussion chapter to justify a line of argument. Therefore, it would be fine with us if this text section remained in the existing chapter.

Reviewer #1: 11) The References cited are appropriate to the topic of the manuscript. Some of the references are not formatted in accordance with the journal requirements, please check the Instructions for authors.

 Authors: We changed the references in accordance to “Instructions for authors”.

Reviewer 2 Report

In the abstract some evidence of the results in terms of value must be added.

Keywords - too much and do not use abbreviations

Introduction.

The first sentence (Line 24-26) has to have some references if there are mentioned numerous publications about several facts - CO2 neutrality, haptics, physical properties etc. The same in sentence 28-31; 45-47.

Line 73 - WPG (Weight Percentage Gain) must be described when the first time mentioned

Results and discussion

More evidence of the results in terms of value must be added. It is not enough to write that there are some difference between the samples. 

For me it is better to understand the results in tables than in the radar diagram. If other reviewers do not mind, better to change them.

Conclusions

In this section there must be your conclusions not from other authors. Therefore the sentences (Line 300-303): "As already known from previous publications, plasma pretreatment causes faster in filtration of liquids (e.g. adhesives) into the wood body. The efficiency of plasma pretreatment may depend on the wood species, the type of wood modification, as well as the type of discharge. " must be moved to the introduction or results section, using references of course.

Line 305 - how many times?

Line 306 - positive or negative influence?

Some evidence of the results in terms of value must be added.

Author Response

Responses to the Reviewer 2 comments and suggestions for corrections

Dear Reviewer, dear Editor,

Thank you very much for the valuable and helpful comments on our submitted manuscript. All changes in the manuscript are highlighted using the "Track Changes" feature.

Reviewer #2: 1) In the abstract some evidence of the results in terms of value must be added.

Authors: We have inserted a sentence in the text showing an example from the mechanical properties: “For example, it was shown that plywood made from plasma-pretreated native beech veneers shows an increase in bending strength of about 8%, and from plasma-pretreated and thermally-modified beech veneers an increase of about 10% compared to the reference.”

Reviewer #2: 2) Keywords - too much and do not use abbreviations

Authors: Thank you for your suggestions. We have replaced the abbreviation "DBD" with "dielectric barrier discharge" and deleted the word "load".

Reviewer #2: 3) The first sentence (Line 24-26) has to have some references if there are mentioned numerous publications about several facts - CO2 neutrality, haptics, physical properties etc. The same in sentence 28-31; 45-47.

Authors: The missing references were inserted into the corresponding text passages.

Reviewer #2: 4) Line 73 - WPG (Weight Percentage Gain) must be described when the first time mentioned

Authors: The abbreviation “WPG” was changed accordingly.

Reviewer #2: 5) More evidence of the results in terms of value must be added. It is not enough to write that there are some difference between the samples.

Authors: Since there is no specific line information in the comment, we assume that the comment refers to the whole chapter. The values for impregnation times, liquid uptake, WPG and thickness swelling were given or can be taken from the graphs. All values for the mechanical properties are shown in the radar diagrams, and the explicit data can be obtained from the two tables including standard deviation and statements on significance in appendices A and B.

Reviewer #2: 6) For me it is better to understand the results in tables than in the radar diagram. If other reviewers do not mind, better to change them.

Authors: Since the loading of the melamine resin was set approximately the same in reference and plasma samples, no large differences between the values were expected. For this reason, we decided to use radar diagrams to show possible differences in mechanical properties in a striking way. The detailed values of the mechanical properties including standard deviation and statements on significance are additionally presented in the tables in appendices A and B.

Reviewer #2: 7) In this section there must be your conclusions not from other authors. Therefore the sentences (Line 300-303): "As already known from previous publications, plasma pretreatment causes faster in filtration of liquids (e.g. adhesives) into the wood body. The efficiency of plasma pretreatment may depend on the wood species, the type of wood modification, as well as the type of discharge. " must be moved to the introduction or results section, using references of course.

Authors: At the suggestion of the reviewer, the two corresponding sentences were removed from the conclusion.

Reviewer #2: 8) Line 305 - how many times?

Authors: The sentence was supplemented accordingly

Reviewer #2: 9) Line 306 - positive or negative influence?

Authors: We have intentionally kept this formulation value-free, since the aim of this study is to show influences of plasma treatment of veneers in general on the performance in terms of mechanical properties of the plywood.

Reviewer #2: 10) Some evidence of the results in terms of value must be added.

Authors: A sentence with exemplary results has been included.

Reviewer 3 Report

In my opinion, the topic of the manuscript is very interesting and the whole work is well prepared. However, the state of the art is not properly described in the introduction section, the most significant findings on the specific research area of plasma treatment and veneers modification are not presented and in parallel, the number of references used in this work is really low. The findings should be correlated to literature findings in discussion section.

Author Response

Responses to the Reviewer 3 comments and suggestions for corrections

Dear Reviewer, dear Editor,

Thank you very much for the valuable and helpful comments on our submitted manuscript. All changes in the manuscript are highlighted using the "Track Changes" feature.

Reviewer #3: 1) In my opinion, the topic of the manuscript is very interesting and the whole work is well prepared. However, the state of the art is not properly described in the introduction section, the most significant findings on the specific research area of plasma treatment and veneers modification are not presented and in parallel, the number of references used in this work is really low. The findings should be correlated to literature findings in discussion section.

Authors: Thank you very much for the valuable comments! Basically, you are absolutely right. If this publication were to be considered stand-alone, then the information (such as the number of references used in this work, state of the art in the introduction, the main results in the specific research field of plasma treatment and modification of veneers, also the discussion of causal relationships in the discussion section) would be insufficiently presented.

However, with this publication we would like to create a direct connection to two already published papers of ours with the same materials and thematic approaches. In the chapters "Introduction" and "Materials and Methods" we wrote (lines 68-78) that the current publication has a direct relation to the two publications of ours mentioned above. Therefore, these noted contents are discussed in detail in these already published publications.

Round 2

Reviewer 3 Report

The authors provided explanations on their manuscript preparation. I would propose to highlight this connection to the 2 previous publications of them in the abstract as well. The manuscript can be published in my opinion as it is.